# Bispectral Index Monitoring and Observer Rating Scale Correlate with Dreaming during Propofol Anesthesia for Gastrointestinal Endoscopies

**DOI:** 10.3390/medicina58010062

**Published:** 2021-12-31

**Authors:** Helena Matus, Slavica Kvolik, Andreja Rakipovic, Vladimir Borzan

**Affiliations:** 1Neuropsychiatric Hospital Dr. Ivan Barbot, 44317 Popovača, Croatia; matushelena@gmail.com; 2Medical Faculty, Department of Anesthesiology and ICU, Osijek University Hospital, J. Huttlera 4, 31000 Osijek, Croatia; 3Department of Anesthesiology and ICU, Osijek University Hospital, J. Huttlera 4, 31000 Osijek, Croatia; andreja.rakipovic@gmail.com; 4Department of Gastroenterology, Osijek University Hospital, J. Huttlera 4, 31000 Osijek, Croatia; borzan.vladimir@kbco.hr

**Keywords:** propofol, anesthesia, general, endoscopy, gastrointestinal, dreams, facial expression, smiling

## Abstract

*Background and objectives*: Dreaming is a commonly reported side effect of propofol anesthesia. *Materials and Methods*: We investigated the inci-dence and character of dreams in patients undergoing intravenous propofol anesthesia and cor-related it with an observer rating scale of facial expression on the seven-point scale from pain to smile. A total of 124 patients undergoing gastrointestinal endoscopy were recruited in the pro-spective observational study. Bispectral index (BIS), blood pressure (BP), and pulse were moni-tored. Upon emergence from anesthesia, the patient’s facial expression was rated numerically. Thereafter, patients were asked whether they had dreams and to rate their dreams as pleasant or unpleasant. The mean age of participants was 53; body mass index, 26.17; duration of procedure, 20 min; and average propofol dose, 265 mg. *Results*: Dreaming was reported by 43% of patients. Dreams were pleasant in all but one patient. There was a significant correlation of the observer’s rating of facial expression with dreaming (r = 0.260; *p* = 0.004). Dreamers had higher scores of observer rating of facial expression (1 (0–2) vs. 0.5 (0–1), *p* = 0.006). *Conclusions*: BIS values were lower in the dreamers vs. non-dreamers 2 min after the endoscopy started (48 (43–62) vs. 59 (45–71), *p* = 0.038). Both BIS and observer ratings correlate with dreaming in patients undergoing gastrointestinal endos-copy. Trial registration number: NCT04235894.

## 1. Introduction

Propofol is an intravenous agent widely used in clinical settings for induction of general anesthesia or sedation. Its use is increasing in ambulatory diagnostic and therapeutic procedures being undertaken in gastrointestinal endoscopy, bronchoscopy, and radiology suites, as well as in ICUs [1]. A meta-analysis of the use of propofol in gastrointestinal endoscopy showed it is safe and effective either alone or in combination with traditional sedatives, including benzodiazepines and opioids [2]. Its use was associated with shorter recovery and discharge periods, higher post-anesthesia recovery scores, better sedation, and greater patient cooperation than traditional sedation, without an increase in cardiopulmonary complications [3].

Short acting time, fast recovery, and few side effects are the advantages of propofol in clinical practice. Unconsciousness occurs 30 s after the 1.5–2.5 mg/kg propofol injection and lasts up to 10 min [4,5]. It is generally recognized to be safe for anesthesia and sedation. Adverse effects commonly include pain after injection, respiratory depression, hypotension, and/or bradycardia. Premedication, underlying cardiopulmonary disease, and advanced age are the usual risk factors for adverse effects [5].

Propofol is known for its pleasant subjective effects after sedation. Patients have reported a broad spectrum of feelings after propofol administration, ranging from a general feeling of well-being and stress relief to elation, euphoria, sexual fantasies, and dreams [6]. This effect is welcomed because it increases the patient’s acceptance of anesthesia and willingness to undergo painful procedures, such as colonoscopy, in the future. Horiuchi et al. showed most patients rate their overall satisfaction with colonoscopy under propofol anesthesia as excellent and are willing to repeat the same procedure [7].

Unfortunately, propofol has become increasingly abused because it is easily accessible, has a rapid onset of action, and has an ultra-short duration of action without any obvious long-term residual side effects [8]. Molecular, animal, and human pharmacological studies strongly suggest the potential for abuse of propofol. Several case reports on human abuse of propofol evidence that it induces dependence and can lead to death [9]. Even though physical dependency is rare, psychological dependency is an important phenomenon. Korea was the first country to classify propofol as a controlled substance in 2011 due to abuse by non-healthcare professionals, mostly young women in their 20s [10]. As suggested by Tezcan et al., pleasant dreams may contribute to propofol’s euphoric effects and potential for abuse [8].

Propofol has been associated with dreams in many studies. These studies have shown different percentages of patients dreaming under propofol anesthesia. Strait et al. reported an incidence of dreaming of 25.5% in patients undergoing colonoscopy. They linked dreams with younger age and higher propofol dose. Patients who had dreams were more satisfied with their care than those who did not [11]. A study carried out by Xu and colleagues showed an incidence of dreaming of 24% and a statistically higher incidence of dreams in men than in women [12], while Tezcan et al. reported an incidence of dreaming of 42%, with no sex differences [8]. All studies agree that dreams under propofol anesthesia have positive emotional content in most cases. Pleasant dreams could contribute to reduced anxiety and overall satisfaction with anesthesia. Gyulaházi et al. proved that perioperative dreams can be increased by preoperative suggestions [13]. They suggest dreams occurring during anesthesia should be turned in a favorable direction by choosing proper induction agents (such as propofol) and through pleasant suggestions during induction of anesthesia, bacause pleasant dreams may contribute to patient satisfaction [13].

The existence of dreams under propofol anesthesia is well known to anesthesiologists. Most patients find these dreams extremely vivid and comfortable. Upon emergence from anesthesia, we often heard the statement, “What you gave me is great!” or “Can you bring me back?” These statements were almost always accompanied by a smile on patients’ faces, suggesting that they were satisfied. Based on these observations, we aimed to quantify our clinical observations.

Observer rating of facial expression is common during emergence from anesthesia but is not commonly registered. A correlation between a subjective observer’s assessment and objective BIS value was found by Bagchi et al. [14]. They concluded that the observer’s assessment of awareness/sedation score and BIS score correlate strongly during recovery from sedation with propofol or midazolam during spinal anesthesia for elective infraumbilical surgeries [14]. Observing facial expressions is part of the frequently used pain rating scales. Pain may be influenced by the procedures themselves, and patients subjected to unpleasant and painful procedures may have low observer rating scores [15]. 

In this study, we investigated whether observer rating score correlates with the presence of dreams under propofol anesthesia for gastrointestinal endoscopies. We also aimed to determine whether there is a difference according to gender, body mass index, ASA status, smoking status, drug use, duration of anesthesia, propofol dose, arterial blood pressure, pulse, and bispectral (BIS) index between dreamers and patients who had no dreams.

## 2. Materials and Methods

After institutional ethics committee (R2:20860-7/2016) approval was obtained, a total of 130 consecutive patients undergoing gastrointestinal endoscopies at Osijek University Hospital, Croatia, EU were included in the prospective observational study from September 2017 to March 2018 (trial registration number NCT04235894). Written informed consent was obtained from all participants. Patients who were younger than 18 years, unscheduled patients who were not hemodynamically stable, those who were not able to understand study protocol, and those who did not sign written informed consent were not included in the study. Patients whose procedure was longer than 60 min and patients with incomplete records were excluded from the study. Ultimately, 124 patients were analyzed (Figure 1). 

A single investigator not providing anesthesia for endoscopic procedures registered all the data. Demographic characteristics, i.e., sex, age, weight, height, comorbidities, ASA status, and preoperative drug consumption were noted before anesthesia data were obtained by preoperative examination at the anesthesiology policlinic. After the patient was admitted to the endoscopy unit, a peripheral venous cannula was inserted, and BP, pulse, and BIS were registered in all patients using Draeger Infinity Delta XL monitors.

Intravenous anesthesia with propofol started with 0.5 mg kg^−1^ and was titrated until the patient was unresponsive to painful stimuli and maintaining spontaneous breathing. Measurement points for BP, pulse, and BIS were registered before anesthesia (time 1), 1 min after the first propofol dose (time 2), immediately after the start of the intervention (time 3), 2 min after the start of the endoscopy (time 4), 5 min after the start of endoscopy (time 5), and finally, after emergence from anesthesia (time 6). /Total propofol dose was registered at the end of anesthesia, as recorded on anesthesia chart. Upon emergence from anesthesia, the patient’s facial expression, reflecting both pain and mood, was rated numerically by the investigator on the seven-point scale [16]. Facial expression upon emergence from anesthesia was rated as very happy if the patient was laughing out loud; happy if the patient had smiling eyes and mouth; a little happy if the patient had smiling eyes or mouth; neutral if the patient exhibited few signs of sadness or dissatisfaction; sad if the patient had as painful grimace; and very sad if the patient exhibited a loud expression of pain [16]. For the purposes of statistical analysis, these facial expressions are numbered as shown in Table 1. 

After the procedure was finished and patients were fully awake prior to being transferred to the recovery room, they were asked whether they had dreams during anesthesia and whether their dreams were pleasant or not.

Statistical analysis was done using SPSS for Windows (IBM SPSS Statistics for Windows, Version 20) and MedCalc^®^ Statistical Software version 20.014 (MedCalc Software Ltd., Ostend, Belgium; https://www.medcalc.org; 2021). Normality of distribution was done using the Kolmogorov-Smirnov test. Comparison between dreamers and non-dreamers was analyzed using χ^2^ or Fisher’s exact test for nominal variables or Mann-Whitney *U* test for continuous data. A correlation between two continuous variables was done using Pearson and between ordinal variables using Spearman correlation. Statistically significant difference between dreamers and non-dreamers was set as *p* < 0.05.

## 3. Results

The median age of 124 patients (48 men and 76 women) undergoing gastrointestinal endoscopies was 56 (42.5–65 yrs); body mass index, 26.17; median duration of procedure, 17 min (14–23.25); and median propofol dose, 240 mg (200–300 mg). A total of 52 patients reported dreaming under anesthesia. Dreams were pleasant in all but one patient. No differences were registered regarding sex, age, ASA status, average preoperative drug consumption, smoking status, duration of anesthesia, or type of procedure (Table 2). Psychiatric diagnoses that may influence dreaming were reported by four patients (5.6%) among 72 non-dreamers and in two patients (3.8%) among 52 dreamers, with no differences between groups (*p* = 1). The use of psychoactive drugs was greater than the number of psychiatric diagnoses recorded. In the non-dreamer group, there were 58 patients (80.5%) who did not use any psychoactive drug, 10 (13.9%) who used one drug—either sedative, hypnotic, antiepileptic, antidepressant, or other psychoactive drugs—and four patients (5.6%) who used two or more drugs, and 39 (75%), 9 (17.3%), and 4 (7.7%) patients in the dreamer group, respectively (*p* = 0.754). Opioid use was also registered. There were eight users of tramadol and its combinations among non-dreamers and four in the group of dreamers (*p* = 0.759).

Statistical analysis confirmed that the only difference between dreamers and non-dreamers was regarding observer rating score for facial expression after anesthesia (*p* = 0.006), with 35 non-dreamers having positive facial expressions.

There were no significant differences in systolic blood pressure between groups at any measurement point. Pulse at T2, T3, and T4 measurement points was lower in the non-dreamers (Figure 2).

Significantly lower BIS values were registered 2 min after the endoscopy started in the dreamers (median 48 vs. 59 in non-dreamers, *p* = 0.037) at T4 point (Figure 3).

Mean observer rating score for facial expression showed positive correlation with dreaming (r = 0.260; *p* = 0.004) but not with age (r = −0.014, *p* = 0.881); BMI (r = 0.089, *p* = 0.324); propofol consumption (r = −0.063, *p* = 0.486); or duration of procedure (r = −0.167, *p* = 0.064). An ROC analysis confirmed observer rating score of facial expression as significant diagnostic tool for the prediction of dreaming under propofol anesthesia, with a cut-off value ≥1 (AUC = 0.646; sensitivity = 50; specificity = 73.1; 95% CI 0.555 to 0.730, Youden index 0.231, *p* = 0.002) (Figure 4).

Weight-adjusted propofol consumption per minute of anesthesia decreases significantly with the patient’s age (r = −0.357, *p* < 0.001) (Figure 5). 

A trend of weight-adjusted propofol consumption per minute of the anesthesia was significantly negatively correlated with patients’ BMI (r = −0.432, *p* < 0.001), as shown in Figure 6.

## 4. Discussion

This study showed that dreamers wake up with significantly more pleasant facial expressions than non-dreamers following propofol anesthesia. This is a common observation in people who emerge from propofol anesthesia; however, to the best of our knowledge, such observations have not been registered to date. We confirmed that the observation of positive facial expressions by observers is significantly associated with dreams. These pleasant dreams have not been observed as often under any other type of anesthesia. Sometimes opioid-naive patients report a feeling of satisfaction before falling asleep after an opioid injection but no subsequent dreams. It is not clear why only propofol has such effects. One explanation could be sought in by examining it physicochemical properties. Propofol is the only anesthetic drug dissolved as oil-in-water emulsion. A recent randomized trial carried out by Nummela and colleagues using nuclear magnetic resonance spectroscopy confirmed that propofol emulsion alters the lipid profile, including lipoproteins, glycerides, and phospholipids, an integral part of all cell membranes [16]. These effects were not observed in the subgroups of healthy volunteers exposed to the equipotent doses of dexmedetomidine or S-ketamine [17]. Due to the high lipid content of the brain, changes in the brain lipid structure and/or function are possible, which may lead to dreams. On the contrary, S-ketamine, with different effects on the human metabolome, has a hallucinogenic effect, which limits its clinical use [18].

Dreaming is usually observed during the REM phase of normal sleep. Both propofol anesthesia and slow-wave NREM sleep stage 3 exhibit similar electrophysiological features, such as the occurrence of prominent slow oscillations and an increase in low-frequency activity [19]. Stefan and colleagues found that broadband slow-wave modulation initially enveloped the posterior cortex when subjects became unconscious but later, when subjects were more deeply anesthetized, encompassed both the frontal and posterior cortices [20]. This could explain the occurrence of dreams in our study patients with lower BIS values, i.e., in somewhat deeper anesthesia, which would probably correspond to the NREM phase. 

In this study, we aimed to indirectly predict the character and appearance of dreams via observer rating scores of facial expressions between very sad and very happy. If we suppose that the patients with more pleasant dreams had higher ORS, they might be more satisfied with anesthesia than non-dreamers. A few studies have been conducted on this topic. A study on 51 patients carried out by Cascella and colleagues did not find any difference in satisfaction with propofol anesthesia between dreamers and non-dreamers [21]. On the contrary, other studies have concluded that dreamers were, on average, more satisfied than non-dreamers [8,11]. The results are inhomogeneous, possibly owing to different study designs, as it is difficult to objectively assess patients’ satisfaction. However, in our study and all those studies carried with propofol, most dreams were pleasant, and dreamers were, on average, at least as happy with anesthesia as non-dreamers. 

A study conducted by Tezcan et al. on 50 patients undergoing colonoscopy under propofol anesthesia showed similar results to those our study [8]. The incidence of dreaming was 42%, just as in our study. Gyulahazi et al. investigated the effect of preoperative suggestions on perioperative dreams in three anesthesia regimes, including propofol-only anesthesia. One day before surgery, the patients in the “dream film group” were instructed to think of a favorite place where they would want to travel during anesthesia and were shown a series of corresponding images at induction [13]. The incidence of spontaneous dreams under propofol anesthesia was 39% in the control group, and in the “dream film group”, the incidence of dreams was 70% [13]. In this way, it is possible to increase the frequency of pleasant dreams by suggestive preparation, perhaps leading to increased satisfaction with anesthesia. Suggestive preparation could thus reduce fear of a stressful and painful endoscopic procedure.

A different ratio of dreamers between men and women under propofol anesthesia hs been observed. In research on 100 men and 100 women undergoing short propofol sedation for upper gastrointestinal endoscopy, Xu et al. showed that men dream statistically more often than women. They also found out that men reported dreams that were more meaningful, familiar, vivid, and memorable, with a higher incidence of recall [12]. A study by Tezcan and colleagues on a smaller sample did not find a statistical difference, although men had a higher dreaming ratio [8], as observed in our study. 

In our study, dreamers had lower BIS values during the endoscopy—the same observation as seen in Eer’s study and in the study conducted by Stait and colleagues [11,22]. In clinical work, it is impossible to confirm sleep stage based on BIS values derived from electroencephalogram (EEG) data, as suggested by the American Academy of Sleep Medicine in 2007 [23]. Based on the results of this study, it seems that a shift from deep sleep to the REM phase is not the only important factor for dreaming, as minimal BIS values were almost the same in both groups. Time spent in deep sleep may also be important [19], as observed in our study.

In addition to perceived pleasant facial expressions, observation of discomfort on the patient’s face is also significant. In our study, there were a total of five patients (4%) with visible discomfort and painful grimace, as observed by ORS. It is likely that the pain caused by intestinal distension exists both at the end of endoscopy and after waking up. It is probable that even more patients than observed by facial expression had significant pain after endoscopies. In a study by Park and colleagues, pain with VAS ≥ 1 was associated with female gender and intestinal decompression. Authors observed pain in 26.6% of patients in the decompression group, with VAS 0.68 ± 1.35, and in 65.5% of patients in the control group, with VAS 2.14 ± 2.15 (*p* < 0.001) [24]. These patients should be interviewed and analgesia-adjusted according to the VAS scale after pain assessment.

Awareness during anesthesia can sometimes be confused with dreams. To avoid this possibility, we used clinical criteria and maintained anesthesia in a state where the patient was unresponsive to painful stimuli. In addition, the depth of anesthesia was recorded by BIS monitoring. According to these data, most patients can be said to have had an adequate depth of anesthesia, although even then, awareness cannot be ruled out, as reported by Cascella and colleagues [25].

A limitation of this study is that preoperative psychological evaluation of patients was not performed, so we recorded the number of self-reported psychiatric diagnoses, other diagnoses, and all medications taken by patients. Statistical analysis showed that the factors we recorded were not associated with occurrence of dreams. It is likely that psychological assessment could provide more data and better dream analysis, but in a fast-tracking endoscopy suite, this was not possible. A significant study improvement would be achieved by psychological analysis that a psychologist could perform during the preoperative examination in the anesthesiology clinic. Consequently, facial expression, which is, by its nature, subjective, was evaluated by only one researcher. The same applies to the dreaming criteria and character of dreams. The dreaming criteria have not been defined by qualitative clustering. As many patients do not want to talk about their dreams, repeated questions on this topic and too many attendant staff during the stressful procedure may result in patient dissatisfaction. Therefore, we asked the patient only to say whether they had dreams and, if so, to rate them as pleasant or unpleasant. Intensity of pain after gastrointestinal endoscopy was not registered in this study and may also be associated with the patient’s facial expression. These limitations may be addressed in future research. 

## 5. Conclusions

In this study, we confirmed that patients who dreamed during propofol anesthesia had a happier facial expression, as registered by observer rating score, than non-dreamers. Dreams under anesthesia may increase patients’ satisfaction and thus compliance with future potentially unpleasant procedures, such as gastrointestinal endoscopy. Future studies should confirm whether patients’ psychological characteristics or comorbid conditions influence the incidence of dreaming.

## Figures and Tables

**Figure 1 medicina-58-00062-f001:**
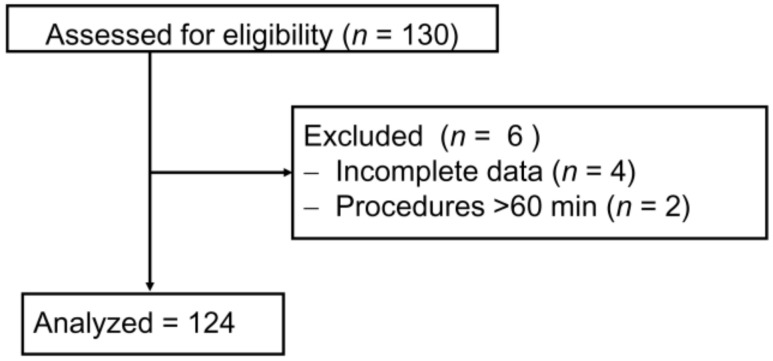
Study flow chart.

**Figure 2 medicina-58-00062-f002:**
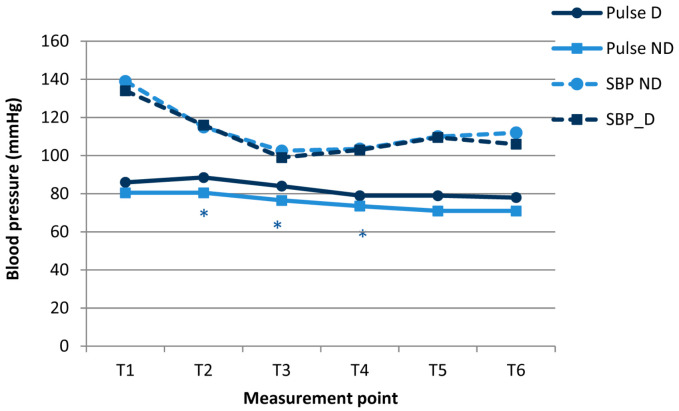
Systolic blood pressure and pulse in the dreamers (D) and non-dreamers (ND) during gastrointestinal endoscopies under propofol anesthesia. * The differences between groups (*p* < 0.05) were confirmed using the Mann-Whitney *U* test.

**Figure 3 medicina-58-00062-f003:**
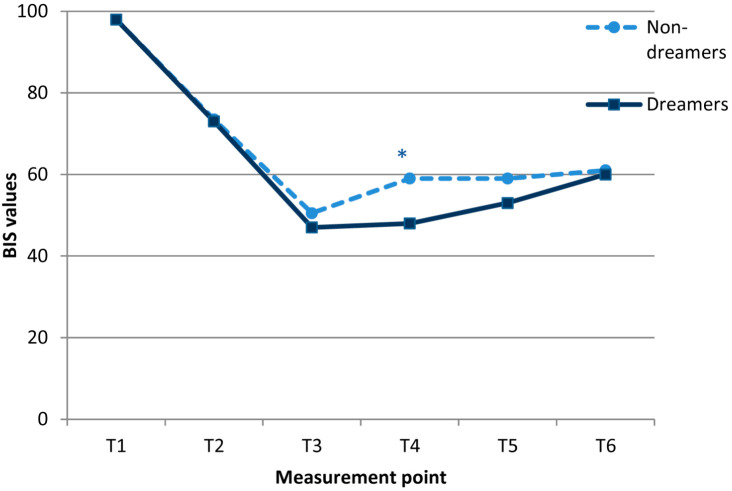
BIS values in the dreamers (D) and non-dreamers (ND) during gastrointestinal endoscopies under propofol anesthesia. * Differences between groups (*p* < 0.05) were calculated using a Mann-Whitney *U* test. Measurement points were before anesthesia (1), 1 min after the first propofol dose (2), immediately after the start of the intervention (3), 2 min after the start of endoscopy (4), five minutes after the start of endoscopy (5), and after emergence from anesthesia (6).

**Figure 4 medicina-58-00062-f004:**
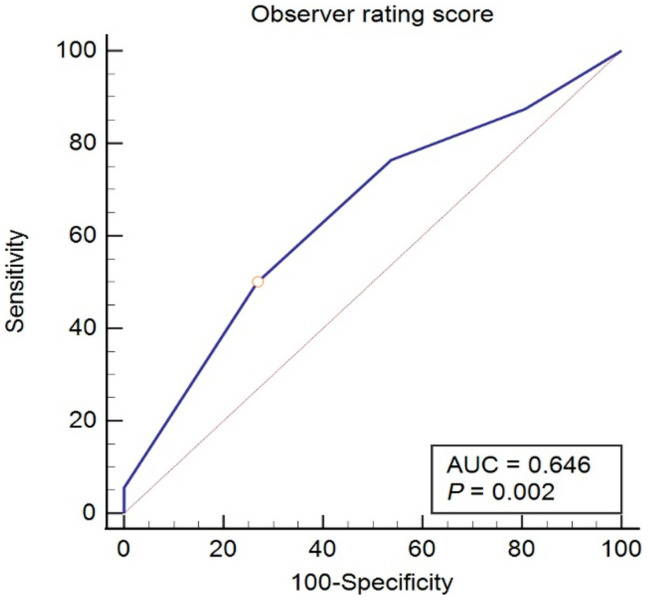
ROC analysis of sensitivity and specificity of observer rating score of facial expression with respect to dreaming or non-dreaming in patients undergoing propofol anesthesia for gastrointestinal endoscopy.

**Figure 5 medicina-58-00062-f005:**
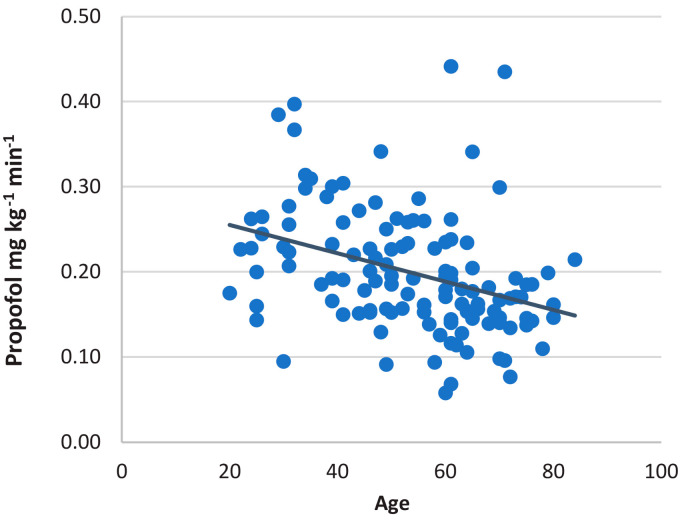
A correlation between weight-adjusted propofol consumption and patients’ age in ambulatory patients undergoing gastrointestinal endoscopies.

**Figure 6 medicina-58-00062-f006:**
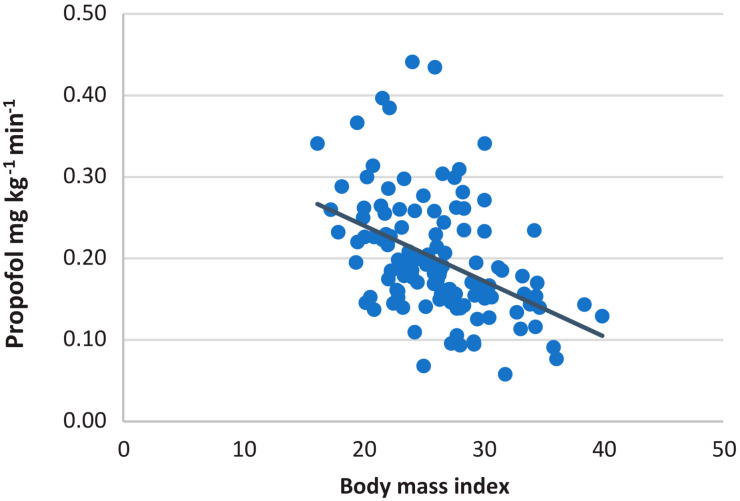
Association of weight-adjusted propofol consumption and patients’ BMI in patients undergoing gastrointestinal endoscopies.

**Table 1 medicina-58-00062-t001:** Observer’s rating scale for facial expression in the patients undergoing gastrointestinal endoscopy.

Numerical Value	Observer’s Rating Scale of Facial Expression
−3	Loud expression of pain, very sad
−2	Painful grimace, sad
−1	Dissatisfaction, little sad
0	Neutral
1	Satisfaction, little happy
2	Smile, happy
3	Laughter, very happy

**Table 2 medicina-58-00062-t002:** Demographic characteristics of dreamers and non-dreamers undergoing ambulatory endoscopic procedures in propofol anesthesia.

	Non-Dreamers (*n* = 72)	Dreamers (*n* = 52)	*p*
Age (yrs ± SD)	57.5	50.5	0.082
Male/female	24/48	24/28	0.191
Body mass index	26.4 (22.9–28.1)	25.6 (22.2–29.5)	0.923
ASA 1/2/3	9/50/13	8/36/8	0.856
Drugs	2 (1–;4)	2 (0.75–;5)	0.967
Smokers/nonsmokers	18/54	16/36	0.543
Colonoscopy/gastroscopy	58/14	43/9	0.819
Duration of anesthesia	16 (13.4–22)	18 (14–25)	0.104
Propofol (mg/kg/min)	0.182 (0.15–0.24)	0.196 (0.15–0.23)	0.367
Observer rating score	0.5 (0–1)	1 (0–2)	0.006

Note: median and interquartile ranges are shown for continuous variables. Statistical analysis was done by the Mann-Whitney test, χ^2^ or Fisher’s exact test.

## Data Availability

All data obtained by this study are presented in the manuscript. Raw data can be obtained from corresponding authors upon reasonable request.

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
