# Peer review of "Bispectral Index Monitoring and Observer Rating Scale Correlate with Dreaming during Propofol Anesthesia for Gastrointestinal Endoscopies"

_medicina, 2021, doi:10.3390/medicina58010062_

Round 1

Reviewer 1 Report

MEDICINA-1475467

Does bispectral index monitoring and observer rating scale correlate with dreaming during propofol anesthesia for gastrointestinal endoscopies.

The study examines incidence of dream under propofol anesthesia.  They examine the correlation with an observer noting facial expression and bispectral index.  The authors suggested a relationship between an observer rating between dreamers and nondreamers plus they noted a lower value in bispectral index of dreamers compared with nondreamers.

Major Concerns

  1. Overall the purpose and impact of the study is not clear. Is there need for an observer to adequately measure a dreamer?  Is it important if propofol has any dreaming effect beyond what is currently known in the literature?  Do the conclusions of the study have a clinical or research benefit?
  2. If it is important for an observer to measure dreaming, I would suggest statistical tests to measure specificity and sensitivity.
  3. Please be more clear on the importance of the BIS values in relation to sleeping and dreaming.
  4. Based on the scope of the study, a figure is missing. In the future, please add in correlation figure between dreaming and facial expression.
  5. The number of observers is not clear. Is this only one observer?  There could be bias with only one.  With multiple observers, also increases the need for additional analysis such inter-rater reliability.
  6. In the future, having a neutral expression as 0 may confound rating dreamers and nondreamers interpretations. Having the extremes of happy and unhappy expressions maybe lost and look similar to neutral nondreamers.

Minor Concerns

  1. Is the x-axis in Figure 1 supposed to match the x-axis in Figure 2.

Author Response

Dear reviewer 1
Thank you for your help and valuable suggestions. We believe our manuscript looks better now. Our answers to your comments and suggestions are in the attached .doc file

Reviewer 2 Report

Dear Editor,

thank you for giving me the opportunity to revise this paper. The manuscript from Helena Matus and co-workers addresses an important topic. Dreaming during sedation or general anesthesia is a fascinating but very difficult to study phenomenon.

The manuscript is very interesting, but some clarifications need to be made. In the introduction, the authors seem to suggest that the propofol-induced dreaming is strictly connected to the propofol-related abuse effect. Previous studies postulated that propofol can mimic the action of additive  substances sucha as  alcohol. It can produce psychotropic effects (healthy volunteers showed that sub-anesthetic doses of propofol provoke feelings of “drunk” along with decreased feelings of “in control of thoughts” and “in control of body”). By the way, this could affect the content of dreams, but it is certainly not the building block of dreaming.

Please, offer a study flow chart for helping readers.

Include EC number (line 88) and country (line 89).

Have been enrolled patients with history of drug abuse, psychiatric illnesses, cognitive impairments, and those who had taken hypnotics?

They suggest that the authors report the limitations in a specific paragraph. This can serve to protect them from criticism. From experience, the subject is very complex. Within the limits, it must be emphasized that a preoperative psychological evaluation was not carried out; that the dreaming criteria have not been defined (qualitative clustering classes); that the assessments were carried out only upon awakening (dreaming can also be reported at a distance); that no distinction was made between dreaming and emergence awareness (Cascella M, Bimonte S, Amruthraj NJ. Awareness during emergence from anesthesia: Features and future research directions. World J Clin Cases. 2020 Jan 26; 8 (2): 245- 254. doi: 10.12998 / wjcc.v8.i2.245. PMID: 32047772; PMCID: PMC7000929.). Underline thast a key problem in all study on the subject is to establish whether the patient’s report can be interpreted as a dream or it is an awareness with recall episode. Moreover, anesthesia awareness is often a well-recognizable phenomenon.

Put more emphasis on your results, in the discussion . Since this topic is rarely the subject of scientific publications, those who are interested in (like the reviewer) want to read a fascinating article.

Author Response

Dear reviewer 1
Thank you for your help and valuable suggestions. We believe our manuscript looks better now. Our answers to your comments and suggestions are in the attached .doc file. "Please see the attachment."

Round 2

Reviewer 1 Report

The study  still has no addressed the previous concern of impact of the study.  Dreaming during propofol anesthesia is clearly presented in the Introduction and shown in the Results.  However, the importance of facial expressions during anesthesia is still not clear and any argument is conjecture.  The authors rebutted that facial expressions are important to understand the level of consciousness.  While that may be true, what is the medical importance the correlation of positive and negative facial features with dreaming?  What does the medical importance on whether they are having a good dream or bad dream?  With that, how will this be addressed in the medical setting?

I understand the concern of having multiple people in the room, but with no gold standard on facial expressions, I still retain my concerns on having one observer.